# Multimodality Approach for Endovascular Left Atrial Appendage Closure: Head-To-Head Comparison among 2D and 3D Echocardiography, Angiography, and Computer Tomography

**DOI:** 10.3390/diagnostics10121103

**Published:** 2020-12-17

**Authors:** Gianpiero Italiano, Anna Maltagliati, Valentina Mantegazza, Laura Fusini, Maria Elisabetta Mancini, Alessio Gasperetti, Denise Brusoni, Francesca Susini, Alberto Formenti, Gianluca Pontone, Gaetano Fassini, Claudio Tondo, Mauro Pepi

**Affiliations:** 1Department of Cardiovascular Imaging, Centro Cardiologico Monzino IRCCS, 20138 Milan, Italy; anna.maltagliati@ccfm.it (A.M.); valentina.mantegazza@ccfm.it (V.M.); laura.fusini@ccfm.it (L.F.); mariaelisabetta.mancini@ccfm.it (M.E.M.); alessio.gasperetti93@gmail.com (A.G.); denise.brusoni@ccfm.it (D.B.); francesca.susini@ccfm.it (F.S.); alberto.formenti@ccfm.it (A.F.); gianluca.pontone@ccfm.it (G.P.); gaetano.fassini@ccfm.it (G.F.); claudio.tondo@ccfm.it (C.T.); mauro.pepi@ccfm.it (M.P.); 2Department of Cardiovascular Sciences and Community Health, University of Milan, 20138 Milan, Italy

**Keywords:** left atrial appendage closure, atrial fibrillation, 3D transoesophageal echocardiography, computed tomography

## Abstract

Background: Percutaneous left atrial appendage closure (LAAC) requires accurate pre- and intraprocedural measurements, and multimodality imaging is an essential tool for guiding the procedure. Two-dimensional (2D TOE) and three-dimensional (3D TOE) transoesophageal echocardiography, cardiac computed tomography (CCT), and conventional cardiac angiography (CCA) are commonly used to evaluate left atrial appendage (LAA) size. However, standardized approaches in measurement methods by different imaging modalities are lacking. The aims of the study were to evaluate the LAA dimension and morphology in patients undergoing LAAC and to compare data obtained by different imaging modalities: 2D and 3D TOE, CCT, and CCA. Methods: A total of 200 patients (mean age 70 ± 8 years, 128 males) were examined by different imaging techniques (161 2D TOE, 103 3D TOE, 98 CCT, and 200 CCA). Patients underwent preoperative CCT and intraoperative 2D and 3D TOE and CCA. Results: A significant correlation was found among all measurements obtained by different modalities. In particular, 3D TOE and CCT measurements were highly correlated with an excellent agreement for the landing zone (LZ) dimensions (LZ diameter: r = 0.87; LAA depth: r = 0.91, *p* < 0.001). Conclusions: Head-to-head comparison among imaging techniques (2D and 3D TOE, CCT, and CCA) showed a good correlation among LZ diameter measurements obtained by different imaging modalities, which is a parameter of paramount importance for the choice of the LAAC device size. LZ diameters and area by 3D TOE had the best correlation with CCT.

## 1. Introduction

Atrial fibrillation (AF) is an independent risk factor for ischemic stroke and thromboembolic events. It significantly increases mortality and morbidity and can cause serious disabilities, as well as longer hospitalization [1,2]. According to the current European Society of Cardiology Guidelines, patients at risk of stroke with contraindications to oral anticoagulant (OAC) or who must interrupt OAC treatment due to major bleeding can be considered for endovascular LAAC [3]. It represents a safe, feasible, and increasingly world-wide treatment, even though the complex LAA morphology might affect the pre-operative evaluation of its anatomical features. Multimodality imaging is an essential tool for LAA closure. Conventional cardiac angiography (CCA), two-dimensional transoesophageal echocardiography (2D TOE) and, especially, three-dimensional transoesophageal echocardiography (3D TOE) have been demonstrated to be valid techniques to assess LAA geometry, in addition to computed tomography (CCT), which is considered to be the gold standard modality to obtain high-quality images and accurate dimensions of the LAA. Few studies have searched for correlations among all different imaging techniques evaluating LAA morphology to establish the best procedural strategy for the placement of LAA occlusion devices [4,5,6,7]. The aim of our study was to evaluate the LAA dimension and morphology in patients undergoing LAA closure and compare data obtained by different imaging modalities (2D TOE, 3D TOE, CCA, and CCT).

## 2. Methods

A total of 200 consecutive patients eligible for LAAC were retrospectively enrolled between March 2010 and September 2020 at Centro Cardiologico Monzino (Milan, Italy). Patients with left ventricular systolic dysfunction (left ventricular ejection fraction < 30%), valvular heart disease, prosthetic heart valve or mitral valve repair, and congenital heart disease were not included. Along with the assessment of LAA thrombosis and shape, measurements of the LAA were carried out by 2D and 3D TOE, CCA, and CCT, and compared between each other. Specifically, the ostium (OS) and the landing zone (LZ) diameters, as well as LAA depth were measured. The longest diameters and depth were then considered for the analyses. The occluder size was decided by interventionists according to the range tables provided by the manufacturer, based on pre- and intraoperative imaging results obtained by 2D TOE and/or CCA. To assess reproducibility of the different measurements, intra-observer and inter-observer variability were evaluated in a subset of 15 randomly selected patients. Each parameter was evaluated ≥4 weeks later on the same data set by the main investigator and by a second experienced observer blinded to the results obtained by the main investigator. All patients were fully informed about the procedure and provided written informed consent. The study was approved by the institution’s human research committee and by the institutional review board. All procedures performed in the study involving human participants were in accordance with the ethical standards of the institutional and/or national research committee and with the 1964 Helsinki declaration and its later amendments or comparable ethical standards.

### 2.1. Conventional Cardiac Angiography 

Angiography (Siemens Artis Zee, Forchhein, Germany) was performed during LAA closure by expert operators. After transeptal puncture, a 5-F marker pigtail was advanced into the LAA, and cineangiograms were performed in multiple projections to ascertain the LAA anatomy and measurements. Caudal projections were used to visualize the mid-distal LAA, whereas right anterior oblique cranial projections were used to visualize the ostium and proximal LAA. The LAA neck and lobes were identified by the use of contrast injections through the sheath from different angulations. The measurements of the ostium, landing zone (at 10 mm), and depth were taken at end-systole. The maximum diameters were recorded (Figure 1A). Measurements were made with angiography-dedicated software.

### 2.2. Transesophageal Echocardiography

In this study, an iE33 and an EPIQ ultrasound system (Philips Medical Systems, Andover, MA, USA) equipped with an X7-2t fully sampled 3D TOE transducer were used to perform 2D and 3D TOE. All images were digitally stored for offline analysis.

2D TOE was performed according to standard clinical protocol with the patient under conscious sedation (e.g., Midazolam 2–5 mg). All measurements were acquired by a standard mid-esophageal window with slight retroflexion (50–70 degree) at 0°, 45°, 90°, and 135°. A full 0°–135° sweep was performed to assess the LAA shape, size, and number of lobes. Maximum (D1) and minimum (D2) diameters of the OS and LZ were obtained. 

The OS measurements were calculated from the origin of the left circumflex artery to the tip of the left superior pulmonary vein, whereas the LZ measurements were calculated 10 mm deeper with respect to the LAA orifice. Finally, the LAA depth was generally measured from the orifice line to the apex of the LAA (Figure 1B). The 2D images were analyzed on-line. 

Real-time 3D TOE was performed, acquiring a pyramidal data set including the entire LAA, using both the zoom mode and the full-volume mode. The frame rate of each image was set at approximately 14–24 volumes per second. Subsequently, pyramidal data sets were cropped along designated *x*-, *y*-, and *z*-axes to remove remaining non-relevant anatomic structures and to improve the visualization of LAA. All images were reviewed online to guarantee adequate 3D visualization of the LAA in the recorded ultrasound images and occasionally repeated over time to ensure optimal image quality. The 3D data sets were imported and analyzed off-line with a dedicated software (QLab-3DQ, Philips Medical Systems) for LAA quantification. The alignment of the LAA long axes in the three different dimensions using the multiplanar reconstruction (MPR) mode allowed the visualization of the LAA orifice in the short axis (Figure 1C). Then, the area of the OS, as well as D1 and D2, was calculated and the largest size was manually chosen.

In the long-axis view, the LAA orifice was determined by two lines, one connected between the mitral valve annulus passing on the left coronary artery and the tip of the left superior pulmonary vein, and the other between the aortic valve annulus and the tip of the lateral ridge of the left pulmonary vein. LAA diameters and the OS area were measured from the short axis view of the LAA orifice. The LZ size was assessed relocating the transversal line 10 mm below the OS. The LAA depth was measured from the midpoint of the OS to the bottom of the primary lobe. The LAAs were categorized into four different morphologies (windsock, chicken wing, cactus, cauliflower) [8]. 

### 2.3. Computed Tomography 

The examinations were performed using a 64-slice Discovery CT 750 HD (GE Healthcare, Milwaukee, WI, USA) in 14 patients, and a 256-slice wide volume coverage CT scanner (Revolution CT; GE Healthcare, Milwaukee, WI, USA) in 84 cases. The imaging volume extended from the proximal brachiocephalic arteries to the upper abdominal aorta in a craniocaudal direction. All examinations were performed using a verbal command, instructing the patient to hold his or her breath after deep inspiration. Scan parameters for 64-slice Discovery CT were 80 or 100 KVp tube voltage, 64 × 0.625 mm slices configuration, 350 ms gantry rotation time, prospective ECG triggering (SnapShot Pulse, GE Healthcare), with an x-ray window of 0, corresponding to an x-ray window of 100 ms scan time at only one distinct end diastolic phase at 75% of the R–R cycle and adaptive statistical iterative reconstruction algorithm (ASIR) at a level of 50%. For the 256-slices CT, the examinations were performed using the following parameters: peak tube voltage, 100 kV; detector collimation: 160 mm using 256 rows by 0.625 mm on *Z* axis. Detector geometry: 256 rows by 832 detection elements per row. High contrast spatial resolution: 0.23 mm. Slice thickness: 0.625; gantry rotation time, 280 ms; prospective triggering; and iterative reconstruction algorithm (ASIR-V; GE Healthcare).

For all patients, CCT data sets were reconstructed and analyzed on a separate workstation (Advantage Workstation Version 4.5, GE Healthcare). The shape and morphology of the LAA were detected and categorized into the current four groups [8]. 

Similar to TOE, OS and LZ diameters and areas, as well as LAA depth, were measured with reconstruction planes (Figure 1D).

### 2.4. Statistical Analysis

Data are presented as the mean ± standard deviation for continuous variables, whereas categorical variables are presented as frequencies and percentages. Linear regression analysis with Pearson’s correlation coefficient was used to evaluate the relationship between 2D and 3D TTE, CCT, and CCA LAA measurements. Bland–Altman analysis was used to assess the inter-technique agreement by calculating the bias (mean difference) and 95% limits of agreement (defined as 1.96 SD around the mean difference). Both intra-observer and inter-observer variability are reported in terms of intraclass correlation coefficients and coefficients of variation (percentages). Moreover, Bland– Altman analysis was applied to evaluate bias and limits of agreement. Statistical analysis was performed using SPSS 25 (SPSS Inc., Chicago, IL, USA). All results were considered significant with a *p*-value < 0.05.

## 3. Results

From March 2010 to September 2020, 200 consecutive patients with non valvular AF were considered suitable for percutaneous LAAC and were enrolled in our study. The mean age of the study population was 70 ± 8 years, CHA_2_DS_2_-VASc score was 3.2 ± 1.3, and HAS-BLED score was 3.1 ± 1.1. Thirty-three patients (17.5%) had a previous stroke or TIA. All baseline clinical characteristics of the population are reported in Table 1. 

Patients were considered a candidate to percutaneous LAA closure in different clinical settings (Table 2): 147 patients (73.4%) had an absolute contraindication to OAC, most frequently due to a history of major bleeding (29.9%); nine patients had bleeding disorders, such as immune thrombocytopenia, hemophilia, Von Willebrand disease, Osler–Weber–Rendu disease; 42.8% of patients had high bleeding risk (had HAS-BLED score ≥ 3); and 22.4% had a thromboembolic event on OAC.

Twelve patients were not implanted because of LAA thrombosis persistence (n = 9) and too large (n = 2) or too small (n = 1) appendage. Four types of devices were used for LAA closure: Amplatzer^TM^ Amulet^TM^ (Abbott, Plymouth, MN, USA) (n = 119; 63%), Watchman^TM^ (Boston Scientific, St. Paul, MN, USA) (n = 41; 22%), Wavecrest^TM^ (Coherex Medical, Biosense Webster Inc., Irvine, CA, USA) (n = 15; 8%), and LAmbre^TM^ (LifeTech Scientific Co. Ltd, Shenzhen, China.) (n = 13; 7%). As documented by intra-operative CCA and 2D TOE, devices were implanted correctly in patients.

Intraprocedural cardiac complications included nine cases of cardiac effusion, two of which required pericardiocentesis, while one patient underwent surgical drainage with concomitant LAA closure. Two cases of device embolization were observed involving, respectively, a 24-mm Watchman^TM^ and a 26-mm LAmbre^TM^ device, both of which were successfully retrieved without clinical consequences.

### Comparison of the LAA Parameters Among 2D and 3D TOE, CCA, and CCT

Multimodality evaluation of the LAA size was performed in all 200 patients. Specifically, 161 patients underwent 2D TOE, 103 patients 3D TOE, 200 patients CCA, and 98 patients CCT. Fifty-five patients underwent all imaging modalities. The diameters of the LZ and depth were collected by all techniques. The average values of the LZ diameters and LAA depth obtained by each technique are presented in Table 3. 

As regards LZ maximal diameters, good correlations were observed among all imaging modalities: r = 0.71 (*p* < 0.001) for CCA vs. CCT (n = 98 patients); r = 0.79 (*p* < 0.001) for 2D TOE vs. CCA (n = 95); r = 0.68 (*p* < 0.001) for 2D TOE vs. CCT (n = 65); r = 0.87 (*p* < 0.001) for 3D TOE vs. CCT (n = 60); r = 0.79 (*p* < 0.001) for 2D TOE vs. 3D TOE (n = 89); and r = 0.82 (*p* < 0.001) for 3D TOE vs. CCA (n = 66). CCT and 3D TOE measurements showed the highest correlation, the lowest average discrepancy, and the narrowest limits of agreement at Bland–Altman analysis (Figure 2).

Analogously, a good correlation between different imaging techniques was obtained for LAA depth: r = 0.86 (*p* < 0.001) for CCA vs. CCT (n = 98); r = 0.72 (*p* < 0.001) for 2D TOE vs. CCA (n = 95); r = 0.80 (*p* < 0.001) for 2D TOE vs. CCT (n = 65); r = 0.91 (*p* < 0.001) for 3D TOE vs. CCT (n = 60); r = 0.78 (*p* < 0.001) for 2D TOE vs. 3D TOE (n = 89); and r = 0.80 (*p* < 0.001) for 3D TOE vs. CCA (n = 66). Similarly to LZ diameters, the highest correlation was documented between CCT and 3D TOE, as well as the lowest bias and limits of agreement (Figure 3).

Concerning OS and LZ areas obtained by 3D TOE and CCT, we observed a good correlation between the two modalities: R = 0.78 (*p* < 0.001) for OS and R = 0.90 (*p* < 0.001) for LZ area. Furthermore, the LZ area measurements were characterized by very low discrepancy and narrow limits of agreement (Figure 4).

Finally, LZ diameters were significantly shorter for 2D TOE and CCA in comparison with 3D TOE and CCT. Greater LZ diameters, LAA depth, and OS and LZ areas were observed by CCT vs. 2D TOE and CCA.

Intra-interobserver variability by CCT imaging was excellent, confirming high reproducibility for both echocardiographic methods and CCT methods. In this regard, 3D methods (3D TOE and CCT) showed the lowest variability. Concerning the variability for 2D TOE, the inter-observer agreements were not optimal, as shown also for other modalities, especially for depth measures (Table 4).

## 4. Discussion

The main findings of our study in a large series of patients undergoing LAAC are: (a) 2D TOE and CCA underestimate LAA main measurements (especially the LZ diameters and area) as compared to 3D TOE and CCT; (b) 3D TOE and CCT show a high correlation between LAA measurements with very low discrepancies; and (c) 3D TOE and CCT may be proposed as the ideal imaging modalities for the preprocedural LAA assessment.

As previously shown in the field of transcatheter aortic valve replacement, the aortic annulus sizing by 2D echocardiography results in systematic underestimation of the true aortic annular diameter due to its elliptical shape [9]. On the contrary, 3D TOE and CCT-based sizing allows a more precise assessment of the annulus and, consequently, a more accurate prothesis size selection [10].

Similarly, although 2D TOE is the method of choice for LAA anatomical assessment [11], several studies have consistently demonstrated that 2D TOE underestimates LZ diameters compared to cardiac CT [4,5,6,7], and our findings are in agreement with these results. A possible explanation for LZ measurement underestimation by 2D echocardiography is the complexity of the LAA structure, which is characterized by an oval shape and accessory lobes. Quantification of the LZ by 2D TOE and CCA is therefore limited by the 2D nature of these imaging modalities. Indeed, 2D TOE and CCA do not adequately allow a complete spatial visualization of the LAA, providing misalignment of the ideal LZ cut-plane. Despite these limitations, the choice of the device size is based mainly on 2D TOE maximal diameter for the most common devices (Watchman^TM^ and Amulet^TM^).

Nucifora et al. [12] suggested that 3D TOE obviates imperfect measurement of LAA size obtained by 2D TOE and jointly with CCT allows a better analysis of LAA dimensions. This was also reported in a study by Shah et al. [13] showing that 2D TOE underestimates the LAA orifice area compared with 3D TOE imaging. Three-dimensional TOE not only is safe and feasible, but also provides more LAA morphological details, such as differentiation of thrombi from pectinate muscles.

The routine integration of pre-procedural CCT into the work-up for LAA closure appears to be of high importance, because it assures a comprehensive evaluation of the LAA and is associated with favorable outcomes in terms of procedural safety [14]. The role of CCT as a potential pre-procedural imaging modality was addressed by few studies [15,16,17,18]. Saw et al. [17] demonstrated that the use of CCT reconstruction allows more precise assessment of LAA morphology, relationships between the LAA and its surrounding structures, and more accurate measurement of the LAA OS and LZ. Furthermore, LAA measurements obtained by CCT are the largest compared with TOE and angiography. However, CCT is associated with non-negligible radiation exposure, cannot be executed when there is renal impairment or an allergy to the contrast agent, and cannot be performed at bedside providing real-time images of the LAA.

According to early clinical experiences with LAA occlusion, an adequate device size tends to be 20–40% larger in diameter than predicted by 2D TOE [18]. Indeed, Al-Kassou et al. [19] showed that the 3D TOE derived diameter from the perimeter of the LZ is the most accurate parameter for determining the optimal device size. Our study reinforces the hypothesis of a tendency of 2D TOE and CCA to underestimate LAA measurements. Measurements obtained by CCT are slightly longer than those obtained with TOE and CCA, as stated in a previous study [20]. In this study, we observed a higher rate of disagreement for intra-observer variability for CCA, caused by a suboptimal definition of LAA borders. This aspect also concerns LZ diameters, and hence could affect device size selection.

The novelty of our study is the inclusion of 3D TOE evaluation in a head-to-head comparison among different imaging modalities; 2D TOE proved to be the method providing the smallest average diameters, whereas CCT is the method with the greatest mean measurements. CCA has an intermediate position between 2D TOE and CCT, while 3D TOE has the best correlation with CCT both for LZ diameters and LAA depth (Table 3). 

Based upon these data, it is reasonable to believe that the incorporation of 3D technology (either 3D TOE or CCT) in the routine morphological LAA assessment can have several advantages in terms of: (a) precise definition of the LAA anatomy; (b) exclusion of thrombi; (c) accurate assessment of the LZ to select the optimal device size overcoming 2D-techniques underestimation; and (d) procedure guidance and monitoring by 3D TOE.

## 5. Conclusions

An accurate anatomical evaluation and correct sizing of the LAA are mandatory to choose the correct device model and size. Because of its complex morphology, LAA sizing using a 3D imaging method should provide greater accuracy than 2D methods. Because of a superior spatial resolution and thanks to multiplanar reconstructions, 3D TOE and CCT provide more precise and reproducible measurements. 

### Study Limitations

The present study has some limitations. Firstly, it is a single center retrospective study. Furthermore, a large series of consecutive patients undergoing LAA closure were enrolled in the study, but only a minority of them underwent all imaging techniques. 

## Figures and Tables

**Figure 1 diagnostics-10-01103-f001:**
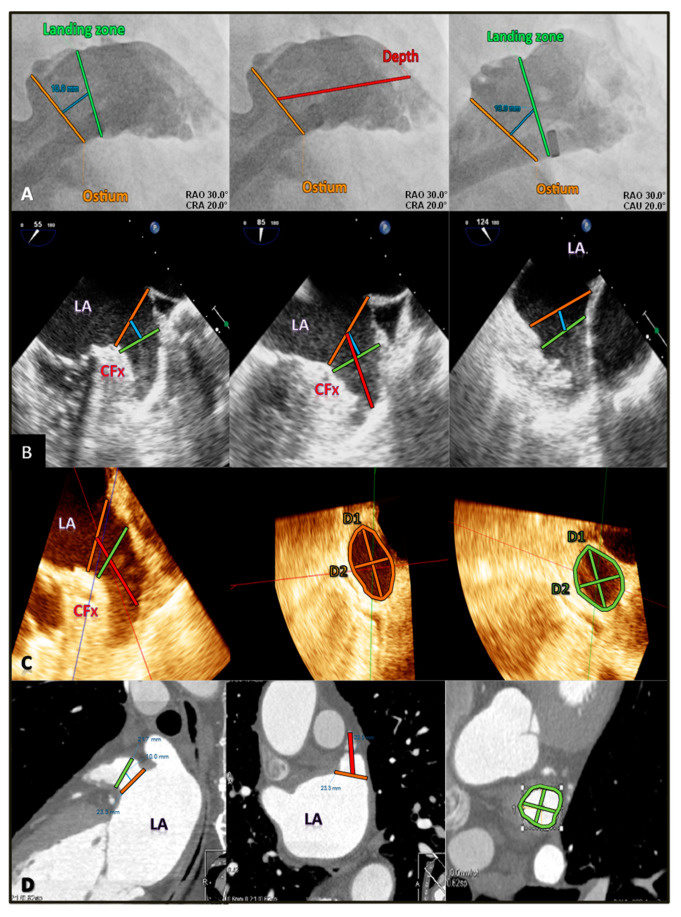
Standard measurements of LAA by different imaging modalities. (**A**) Selective angiography measurements at the echocardiographic ostium (orange line), landing zone (green line) at 10 mm (blue line) from ostium, and LAA depth (red line). (**B**) 2D TOE measurements of the ostium, landing zone, and depth. (**C**) 3D TOE standard measurements of ostium diameters and area, of landing zone diameters and area, and of LAA depth. (**D**) CCT multiplanar reconstruction images with measurements of the landing zone area, maximum and minimum diameter, and LAA depth. LA = left atrium; CFx = circumflex artery; D1 = maximum diameter; D2 = minimum diameter.

**Figure 2 diagnostics-10-01103-f002:**
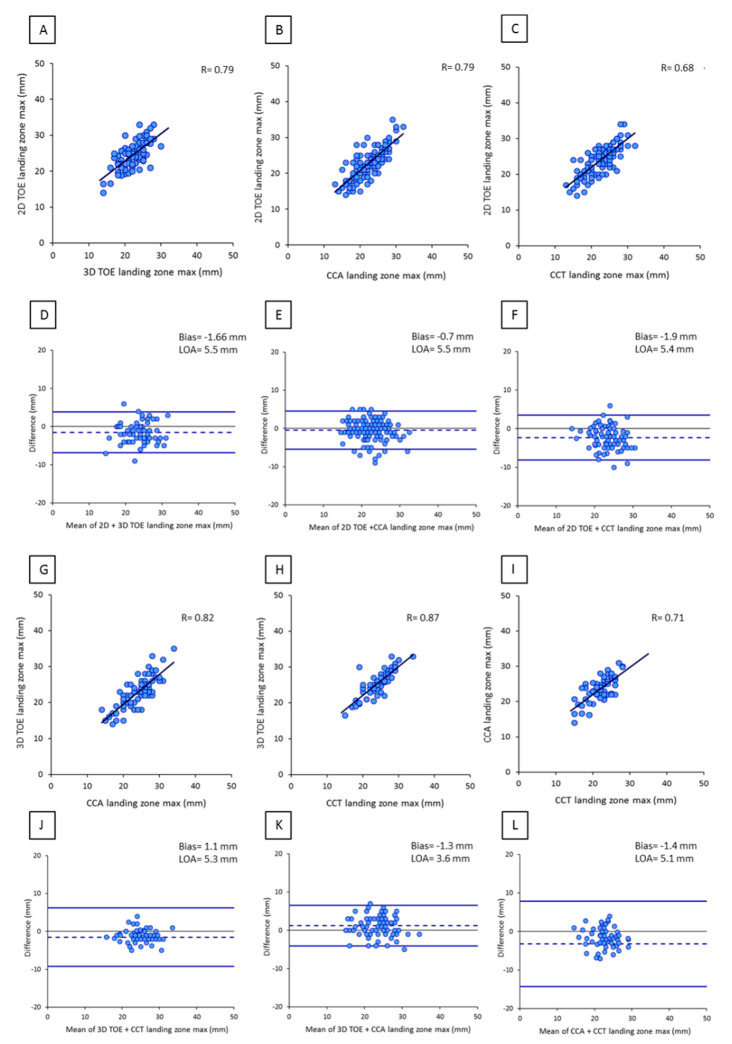
Correlation and agreement between landing zone measurements obtained by 2D TOE, 3D TOE, CCA, and CCT: (**A**) scatter plot of 2D TOE versus 3D TOE measurements, (**B**) scatter plot of 2D TOE versus CCA, (**C**) scatter plot of 2D TOE versus CCT, (**D**) Bland-Altman plot of 2D TOE versus 3D TOE, (**E**) Bland-Altman plot of 2D TOE versus CCA, (**F**) Bland-Altman plot of 2D TOE versus CCT, (**G**) scatter plot of 3D TOE versus CCA measurements, (**H**) scatter plot of 3D TOE versus CCT measurements, (**I**) scatter plot of CCA versus CCT measurements. (**J**) Bland-Altman plot of 3D TOE versus CCA measurements, (**K**) Bland-Altman plot of 3D TOE versus CCT, and (**L**) Bland-Altman plot of CCA versus CCT.

**Figure 3 diagnostics-10-01103-f003:**
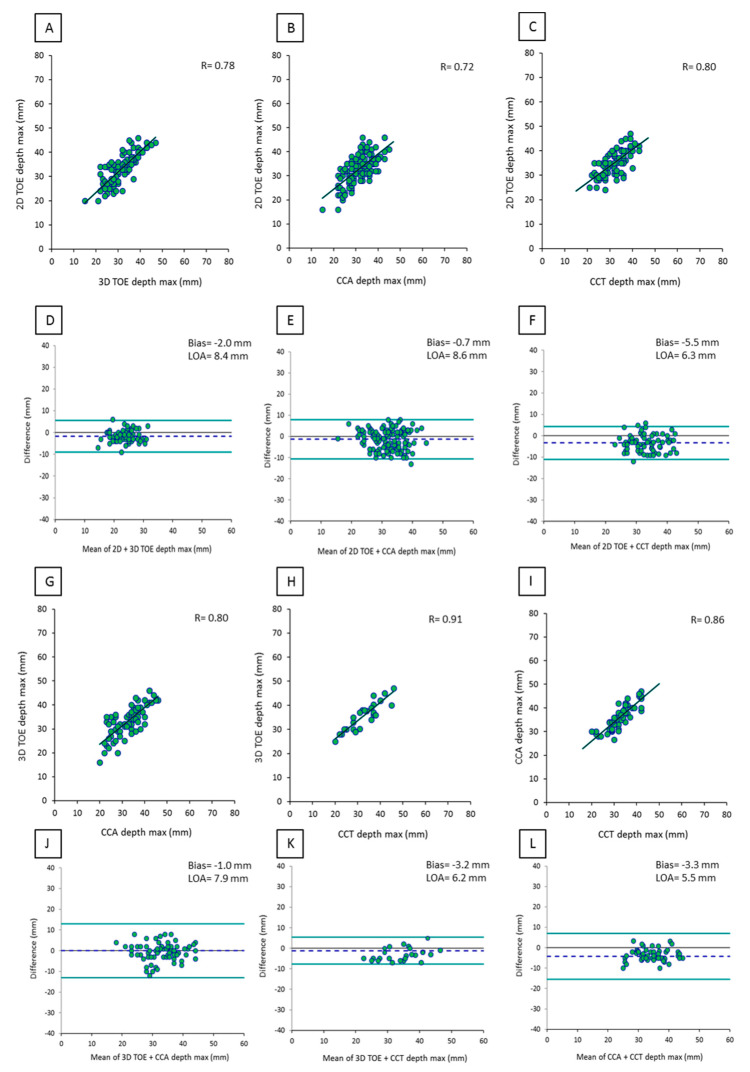
Correlation and agreement between LAA depth measurements obtained by 2D TOE, 3D TOE, CCA, and CCT: (**A**) scatter plot of 2D TOE versus 3D TOE measurements, (**B**) scatter plot of 2D TOE versus CCA, (**C**) scatter plot of 2D TOE versus CCT, (**D**) Bland-Altman plot of 2D TOE versus 3D TOE, (**E**) Bland-Altman plot of 2D TOE versus CCA, (**F**) Bland-Altman plot of 2D TOE versus CCT, (**G**) scatter plot of 3D TOE versus CCA measurements, (**H**) scatter plot of 3D TOE versus CCT measurements, (**I**) scatter plot of CCA versus CCT measurements, (**J**) Bland-Altman plot of 3D TOE versus CCA measurements, (**K**) Bland-Altman plot of 3D TOE versus CCT, and (**L**) Bland-Altman plot of CCA versus CCT.

**Figure 4 diagnostics-10-01103-f004:**
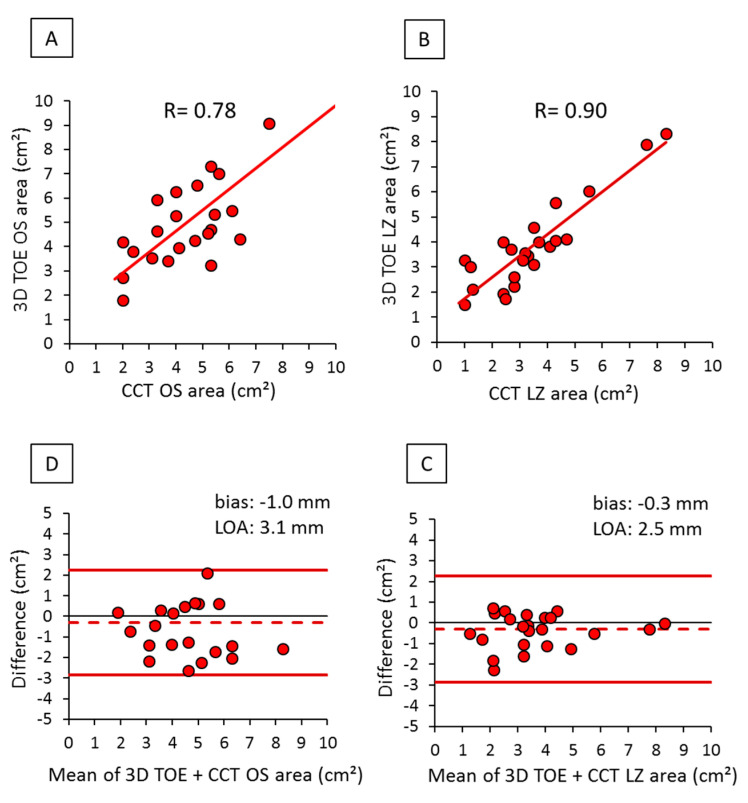
Correlation and agreement between OS and LZ area measurements obtained by 3D TOE and CCT: (**A**) scatter plot of OS areas, (**B**) scatter plot of LZ areas, (**C**) Bland-Altman plot of OS areas, and (**D**) Bland-Altman plot of LZ areas.

**Table 1 diagnostics-10-01103-t001:** Baseline clinical characteristics of the study population.

**Age (years)**	70.3 ± 8.3
**Male, n (%)**	128 (64.1%)
**Body surface area (m^2^)**	1.85 ± 0.20
**Hypertension, n (%)**	158 (78.9%)
**Diabetes mellitus, n (%)**	20 (10.2%)
**Hyperlipidaemia, n (%)**	84 (42.1%)
**Prior stroke/TIA, n (%)**	35 (17.6%)
**Coronary Heart Disease, n (%)**	37 (18.3%)
**Congestive heart failure, n (%)**	27 (13.7%)
**Chronic Kidney Disease, n (%)**	24 (12.2%)
**Contraindications to anticoagulation, n (%)**	147 (73.4%)
**Paroxysmal AF, n (%)**	57 (28.5%)
**Persistent AF, n (%)**	77 (38.7%)
**Permanent AF, n (%)**	65 (32.8%)
**HAS-BLED score**	3.19 ± 1.1
**CHA_2_DS_2-_VASc score**	3.20 ± 1.4

CHA_2_DS_2_-VASc = congestive heart failure, hypertension, age > 75 years, diabetes mellitus, prior stroke or transient ischemic attack or thromboembolism, vascular disease, age 65 to 74 years, sex category; HAS-BLED = hypertension, abnormal renal and liver function, stroke, bleeding, labile international normalized ratio, elderly, drugs, or alcohol.

**Table 2 diagnostics-10-01103-t002:** Clinical reasons for percutaneous LAA closure.

**Previous major bleeding, n (%)**	60 (29.9%)
**Previous minor bleeding, n (%)**	49 (24.4%)
**Thromboembolism on OAC, n (%)**	45 (22.4%)
**Hematologic pathology, n (%)**	9 (4.5%)
**Labile INR, n (%)**	6 (2.7%)
**High HAS-BLED score (≥ 3), n (%)**	85 (42.8%)

OAC = oral anti-coagulant, INR = international normalized ratio.

**Table 3 diagnostics-10-01103-t003:** Landing zone diameter and area, ostium area, and left atrial appendage depth expressed as mean ± standard deviation, measured with different imaging techniques.

	CCA	2D TOE	3D TOE	CCT
**LZ D1 (mm)**	22.46 ± 4.32	21.82 ± 4.09 *	23.75 ± 5.73 *†	24.05 ± 3.39 *†‡
**LAA depth (mm)**	33.95 ± 6.10	30.86 ± 5.73 *	32.79 ± 6.61 †	36.93 ± 6.15 *†‡
**OS area (cm^2^)**			4.68 ± 2.1	4.68 ± 1.7
**LZ area (cm^2^)**			3.29 ± 1.6	3.31 ± 1.27

D1 = maximum diameter; LAA = left atrial appendage; LZ = landing zone; OS = ostium. * *p* < 0.05 vs. CCA; † *p* < 0.05 vs. 2DTOE; ‡ *p* < 0.05 vs. 3D TOE.

**Table 4 diagnostics-10-01103-t004:** Results of the intra- and inter-observer variability on the agreement between repeated measurements of 2D and 3D TOE, CCA, and CCTA images obtained in a subset of 15 randomly selected patients.

		INTRA-OBSERVER			INTER-OBSERVER		
		CV	ICC	Bias ± LOA	CV	ICC	Bias ± LOA
2D	Ostium	8.2	0.732	0.1 ± 7.2	10.0	0.661	1.7 ± 9.0
	Landing zone	8.2	0.675	0.2 ± 6.7	7.8	0.754	0.3 ± 7.0
	Depth	9.9	0.750	−0.8 ± 11.2	13.9	0.584	−5.0 ± 17.3
3D	Ostium	6.4	0.872	0.7 ± 5.9	3.1	0.978	0.9 ± 2.3
	Landing zone	6.0	0.917	0.7 ± 4.4	8.2	0.830	2.0 ± 6.1
	Depth	5.5	0.904	0.4 ± 5.3	8.4	0.849	2.5 ± 6.8
CCA	Ostium	6.3	0.959	0.9 ± 4.2	4.3	0.980	−0.2 ± 3.0
	Landing zone	5.2	0.981	1.3 ± 2.7	2.3	0.988	0.4 ± 2.1
	Depth	4.5	0.928	1.6 ± 4.4	3.1	0.951	0.0 ± 3.7
CCTA	Ostium	1.6	0.992	0.3 ± 1.2	3.5	0.953	0.4 ± 3.1
	Landing zone	1.1	0.995	−0.2 ± 0.7	3.0	0.977	−0.7 ± 1.6
	Depth	1.1	0.994	0.3 ± 1.6	2.7	0.966	−0.1 ± 3.8

CV = coefficient of variability; ICC = interclass correlation; LOA = limits of agreement.

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
