# Peer review of "Multimodality Approach for Endovascular Left Atrial Appendage Closure: Head-To-Head Comparison among 2D and 3D Echocardiography, Angiography, and Computer Tomography"

_diagnostics, 2020, doi:10.3390/diagnostics10121103_

Round 1
Reviewer 1 Report
Interesting study regarding comparison of various imaging methods for LAA closure. The study is well designed and presented.
I have only minor suggestions to make.
-In the abstract the authors mention “..intraoperative 2D and 3D TOE and CCA”. However in the methods section you mention “..preoperative measurements of the LAA were carried out by 2D and 3D TOE, CCA and CCT”. Please clarify to avoid discrepancies between abstract and methods section content.
-Does the extent of LA remodeling/enlargement affect the degree of correlation between 3D TOE and CCT measurements?
-Please check whether references are correctly cited in the main text (for example references 12,13).
Author Response
Response to reviewer #1. Comments
Point 1. In the abstract the authors mention “..intraoperative 2D and 3D TOE and CCA”. However in the methods section you mention “..preoperative measurements of the LAA were carried out by 2D and 3D TOE, CCA and CCT”. Please clarify to avoid discrepancies between abstract and methods section content.
Response 1: Thanks for the suggestion. We would say that 2D and 3D TOE were performed during LAA procedure as well as CCA, while CCT were realized some days before procedures (max 3 days). In the new version, this point will be clarified.
Point 2: Does the extent of LA remodeling/enlargement affect the degree of correlation between 3D TOE and CCT measurements?
Response 2: LA remodeling/enlargement could affect the correlation between 3D TOE and CCT measurements for example as result of hemodynamic changes. For these reasons, we performed CCT only few days before procedure without intravenous liquids intake. Catheter ablation before LAA closure might be a bias for LAA intraoperative and preoperative measures: multicentric registries support the feasibility and safety of performing combined procedures but no data are available about changes in LAA dimensions. In our experience, we have not observed significant LZ variations in the few patients who underwent catheter ablation before LAA closure, but this aspect was beyond the purpose of our study.
Point 3. Please check whether references are correctly cited in the main text (for example references 12,13). Response 3: We forgot reference at number 7 (Wunderlich NC, Beigel R, Swaans MJ, Ho SY, Siegel RJ. Percutaneous interventions for left atrial appendage exclusion: options, assessment, and imaging using 2D and 3D echocardiography. JACC Cardiovasc Imaging. 2015 Apr;8(4):472-488) that caused mess up. Sorry for the inconvenience. Accordingly, we will modify it in the new version.
Reviewer 2 Report
This paper compares various left atrial appendage (LAA) size measurements by different imaging modalities; namely, 2D and 3D trans oesophageal echocardiography (2D and 3D TOE), computed tomography (CCT) and conventional cardiac angiography (CCA). In a total of 200 patients, the results show an good correlation among all measurements obtained from different modalities. The authors conclude the landing zone dimensions as measured by 3D TOE and CCT have the best correlation, as well as the most precise and reproducible measurements.
Overall, the paper is well written and has shown convincing results which agree with existing literatures. There are some minor concerns and comments that need to be addressed.
* Section 2: Please clarify if this study has been approved by an institutional review board.
* Section 2.1: Please describe the angiography imaging system and measurement software used for this study.
* Section 2.2: Please list how many patients went through 2D and 3D TOE scans.
* Section 2.3: There are 14 patients with 64-slice CT scan and 57 with 256-slice scan. How would these add up to the total of 98 CCT scans?
* Page-4, Line 105: “(QLab-3DQ, Philips Medical Systems)” Redundant description. This has already been mentioned earlier.
* Section 3.1: How many patients have been through all modalities?
* Figure-1: Please illustrate what the blue line represents.
* Page 9, Line 223: “Greater LZ diameters, LAA depth and OS and LZ areas were observed by CCT vs. 2D TOE and CCA”. Are their differences significant?
* Page 9, Line 225: “Landing zone measurements by 2D and 3D TTE showed excellent intraobserver and interobserver agreement.” 1.) Please change TTE to TOE. 2.) 2D TOE doesn’t show excellent intra and interobserver agreements. In fact, all measurements (including LZ) in 2D TOE have a higher variability (i.e. higher CV, and lower ICC) than 3D TOE, CCA, and CCT from Table-4. This needs to be mentioned.
* Table-4: There is a greater intraobserver variability than interobserver variability in many of the LAA measurements; particularly in the CCA modality. Please explain the likely causes of a higher inconsistency in the intraobserver measurements.
* Please compare the correlation between 1D line measurements (LZ D1, LAA depth) and 2D area measurements (OS area and LZ area) and see which 1D might correlate the best with the 2D.
* Page 10, Line 282 and 283: “(e) reliable prediction of procedural outcomes and (f) postprocedural surveillance.” Both are not shown in this study as there is no outcome data included.
Author Response
Response to reviewer #2. Comments
Point 1. Please clarify if this study has been approved by an institutional review board
Response 1: The study has been approved by the institution’s human research committee and by the institutional review board. We will insert this point in the new version.
Point 2: Please describe the angiography imaging system and measurement software used for this study.
Response 2: The software used for angiography is Siemens Artis Zee (Forchhein, Germany). We will insert the angiography system and dedicated software in the new version.
Point 3. Section 2.2: Please list how many patients went through 2D and 3D TOE scans.
Response 3: 161 patients underwent 2D TOE and 103 patients 3D TOE, respectively. These data have been reported in section 3 lines 176-177.
Point 4. Section 2.3: There are 14 patients with 64-slice CT scan and 57 with 256-slice scan. How would these add up to the total of 98 CCT scans?
Response 4: Thanks for this observation. We erroneously reported these numbers that will be corrected in the new version: 14 patients with 64-slice CT scan and 84 with 256-slice scan.
Point 5. Page-4, Line 105: “(QLab-3DQ, Philips Medical Systems)” Redundant description. This has already been mentioned earlier.
Response 5: Thanks, we agree that is a redundant data and we will delete technical details of the system in the new version.
Point 6. Section 3.1: How many patients have been through all modalities?
Response 6: Fifty-five patients underwent all imaging modalities (reported in line 177 and 178)
Point 7. Page 9, Line 223: “Greater LZ diameters, LAA depth and OS and LZ areas were observed by CCT vs. 2D TOE and CCA”. Are their differences significant?
Response 7: Thank you for this observation, we found significant difference between LZ D1 CCT vs LZ D1 CCA, LZ 2DTOE and LZ 3DTOE and between DEPTH CCT vs DEPTH CCA, DEPTH 2DTOE and DEPTH 3DTOE. We will modify Table 3 with this statistical addendum.
Point 8. Page 9, Line 225: “Landing zone measurements by 2D and 3D TTE showed excellent intraobserver and interobserver agreement.” 1.) Please change TTE to TOE. 2.) 2D TOE doesn’t show excellent intra and interobserver agreements. In fact, all measurements (including LZ) in 2D TOE have a higher variability (i.e. higher CV, and lower ICC) than 3D TOE, CCA, and CCT from Table-4. This needs to be mentioned.
Response 8: We agree to these observations and we will review our text. As concerns variability for 2D TOE the inter-observer agreements was not optimal as shown also for others modalities. Especially for depth measures, we have observed a higher CV due to great LAA anatomical variability. A new sentence will be added regarding this point.
Point 9. Table-4: There is a greater intraobserver variability than interobserver variability in many of the LAA measurements; particularly in the CCA modality. Please explain the likely causes of a higher inconsistency in the intraobserver measurements.
Response 9: A higher rate of disagreement for intra-observer variability could be caused by a suboptimal definition of LAA borders of CCA, even for landing zone. Furthermore, angiographic loop selection for measurements could be associated with this greater variability. We will add a comment on this point in the discussion.
Point 10. Please compare the correlation between 1D line measurements (LZ D1, LAA depth) and 2D area measurements (OS area and LZ area) and see which 1D might correlate the best with the 2D.
Response 10: The statistical analysis shows that LZ D1 significantly correlates with LZ area (p<0.001), the others correlations (LZ D1 vs OS area, LAA depth with both OS area and LZ area) do not significantly correlate.
Point 11. Page 10, Line 282 and 283: “(e) reliable prediction of procedural outcomes and (f) postprocedural surveillance.” Both are not shown in this study as there is no outcome data included.
Response 11: We agree that out data do not support point e) and f ) of our last sentence. Therefore, these 2 point will be deleted.